# Respiratory Function in Friedreich’s Ataxia

**DOI:** 10.3390/children9091319

**Published:** 2022-08-29

**Authors:** Elena Vinante, Elena Colombo, Gabriella Paparella, Michela Martinuzzi, Andrea Martinuzzi

**Affiliations:** 1IRCCS “E. Medea”, Department of Neurorehabilitation, 31015 Conegliano, Italy; 2Sandwell and West Birmingham NHS Trust, Birmingham G168GX, UK

**Keywords:** Friedreich’s ataxia, respiratory function

## Abstract

Background: Friedreich’s ataxia is an inherited, rare, progressive disorder of children and young adults. It is characterized by ataxia, loss of gait, scoliosis, cardiomyopathy, dysarthria and dysphagia, with reduced life expectancy. Alterations of respiratory dynamics and parameters are frequently observed. However, in the literature there are few, dated studies with small cohorts. Our study aims to make an objective analysis of the respiratory condition of both early and late stage FRDA patients, looking for correlations with the motor, skeletal, speech and genetic aspects of this condition. Materials and methods: This retrospective observational study is based on the collection of clinical and instrumental respiratory data of 44 subjects between 13 and 51 years attending a tertiary rehabilitation centre in northern Italy. The analysis was carried out using Pearson’s correlation test, ANOVA test and post hoc tests. Results: Data show the presence of a recurrent pattern of respiratory dysfunction of a restrictive type, with reduction in forced vital capacity and of flow and pressure parameters. The severity of the respiratory condition correlates with the disease severity (measured with disease-specific scales), with pneumophonic alterations and with the severity of the thoracic scoliotic curve. Conclusions: Respiratory function is impaired at various degrees in FRDA. The complex condition of inco-ordination and hyposthenia in FRDA affects daytime and night-time respiratory efficiency. We believe that the respiratory deficit and the inefficiency of cough are indeed a clinical problem deserving consideration, especially in the context of the concomitant postural difficulty and the possible presence of dysphagia. Therefore, the rehabilitation project for the subject with FRDA should also consider the respiratory function.

## 1. Introduction

Friedreich’s ataxia (FRDA) is a rare autosomal recessive condition, mainly presenting in children and young adults, leading to progressive neuromuscular compromise, loss of ambulation and premature death with no known cure. It is the most common inherited cause of ataxia in Europe, affecting 1 in 29,000 people and usually manifests between 9 and 20 years old, although early and late onset forms also exist [1].

FRDA is caused by an alteration in the frataxin gene, secondary to an unstable GAA repeat expansion. The length of the GAA repeat inversely correlates with age at onset and severity of disease [2]. This leads to reduction in frataxin production which is normally involved in iron metabolism and ATP production. The most affected tissues are the central and peripheral nervous system, the myocardium and the endocrine system.

Clinical features include ataxia, sensory neuropathy, dysarthria and pyramidal weakness leading to increased instability, loss of proprioception, co-ordination and lower limbs reflexes with progressive loss of ambulation on average 15 years after symptoms onset. FRDA is also associated with dysphagia and dysarthria. The neurological symptoms are a consequence of spinocerebellar, corticospinal, cerebellar and sensory nerve pathology. FRDA is associated with endocrine and musculoskeletal abnormalities, cardiomyopathy (the most common cause of death in these patients) and osteopenia [2]. 

Respiratory impairment is not widely documented in patients with FRDA but on the basis of our center experience, we have frequently witnessed respiratory difficulties associated with abnormal spirometry and saturation values in these patients. We have noted, at times, desaturations in the 6-minute walk test, increased fatigability and dysarthria as well as day-time sleepiness and episodes of obstructive sleep apnea (OSA). These are not isolated findings, as shown by Corben et al., who reported the presence of OSA in 21% of patients with FRDA, often associated with nocturnal hypoventilation and an overnight desaturation index (ODI) above 5/h. The study hypothesized that these were caused by an airway collapse, secondary to deficits in pharyngeal musculature [3].

Studies conducted by Cisneros and Braun hypothesized the relationship between respiratory and phonation pathologies [4] and showed how respiratory insufficiency and speech disorders were associated with compromise of the respiratory centers via pontine and medullary atrophy. The ability to formulate speech depends on co-ordination in the abdominal, intercostal, diaphragm and laryngeal adductors muscles. Falker et al. conducted spirometry assessments during syllabic repetition and 12 patients showed paradoxical movements, reduction in vital capacity and neuromuscular respiratory insufficiency. Some of these patients exhibited saccadic movements at the end of respiratory effort, and shortness of breath was noted in expiration, secondary to lack of control on expiratory effort due to absence of co-ordination in respiratory musculature [5].

Scoliosis is frequently encountered in patients with FRDA. In a recent study performed on more than 1000 patients with a mean follow up of 17 years, authors showed how patients with early onset and typical onset FRDA have a higher prevalence of scoliosis at diagnosis (69–80%) when compared to late onset forms (28%) [6]. At diagnosis, scoliosis is the second most common symptom and among non-neurological symptoms, the first. Progression is quicker in growth phases such as puberty, and then slow afterwards. When compared to idiopathic scoliosis, FRDA patients develop more severe and unusual forms such as left convex curves or double curves. The length of GAA1 triplets correlates with the progression of the curve and an early surgical intervention [7].

In this study we systematically collected data with the aim of providing a picture of the respiratory function of patients with FRDA in order to identify whether these patients present significant abnormalities in spirometry and respiratory impairment. We also wanted to identify which factors contribute to worsening respiratory function in patients with FRDA who present with skeletal, muscular and co-ordination abnormalities to identify potential targets of rehabilitation interventions that can improve functional and clinical outcomes.

## 2. Materials and Methods

The study was conducted in accordance with the Declaration of Helsinki and was approved by the competent Ethics Committee (protocol code 548CE/Medea, 7 March 2019). Informed written consent was obtained from all subjects involved in the study.

We retrospectively collected data on clinical and functional status of 44 patients with FRDA between the years 2018–2022. Patients included in the study had to be under the care of IRCCS Eugenio Medea “La Nostra Famiglia” in Conegliano-Pieve di Soligo, Italy, have to have had at least one clinical assessment in this center and have a formal genetic diagnosis of FRDA.

Indicators included in the study were: demographics (sex and age), age of symptoms onset, diagnosis, duration of illness and genetic information regarding GAA triplets, which were collected from patient medical records.

For the respiratory assessment, spirometry was recorded sitting and supine on each patient. Nocturnal oximetry studies were carried out with a pulse oxymeter in all cases except one who was using nocturnal non- invasive ventilation. Capillary blood gas analysis was also performed. The portable spirometry Pony FX Cosmed was used and for the sitting position mobility aids such as wheelchairs or chairs were used. Tests were performed by a single specialized respiratory physiotherapist. The following parameters were recorded: FVC (forced vital capacity), FEV1 (forced expiratory volume in 1 s), TI (Tiffeneau index), MIP (maximal inspiratory pressure), MEP (maximal expiratory pressure), PCF (peak cough flow), PEF (peak expiratory flow. MIP and MEP were measured with micro RPM (MicroRPM, Care Fusion, Hoechberg, Germany) and PCF and PEF with PFM20 (Omron, Essex, UK).

For each parameter, three readings were taken and the best one recorded. In some cases, tests were repeated with and without lumbar support. All the evaluations were performed in accordance with the ATS/ETS recommendations [8].

For all the participants, clinical data on motor function, speech and radiological assessment for degree of scoliosis (with Cobb angle) were also included, as recorded by specialist physiotherapists and radiologists. To ease statistical analysis, patients were clustered in five groups: group 1: <20°, group 2: 20–30°, group 3: 30–40°, group 4: 40–50° and group 5: >50°.

For the motor assessment of these patients, abdominal strength, upper limb co-ordination, disease stage as measured with FARS (Friedreich ataxia rating scale), abdominal force as per MRC (Medical Research Council) score and SARA (scale for assessment and rating of ataxia) were measured.

Motor assessment was carried out by a specialist physiotherapist looking at abdominal strength as measured with MRC, co-ordination, SARA score and disease stage as FARS. SARA scale has scores between 0 = normal and 40 = absolute gravity.

FARS scale was used for the following items:Functional staging (0 = normal, 6 = maximum disability).Upper limb co-ordination (0 = normal, 36 = maximum disability).Upright stability (0 = normal, 28 = maximum disability).

MRC scale with score between 0 = no muscle contraction and 5 = normal strength.

For speech assessment, duration of phonation (number of syllables per expiratory phase and length of vocalization per second), speed of speech (numbers of syllables per second) and duration of expiration were included. The quantitative measures were obtained thanks to PC recording and analysis by a dedicated software (http://www.praat.org/ accessed on 9 August 2022). All patients underwent an assessment by a speech and language therapist, as per local protocol where these values were collected.

Demographics characteristics and clinical data of participants were shown using descriptive statistics. Mean (standard deviation) or median (range) were used for continuous data following normal and non-normal distribution, respectively, and number (percentage) was used for categorical variables. Spearman’s correlation coefficients between respiratory measures and clinical outcomes (motor function, speech and degree of scoliosis’ measures) were computed. One-way ANOVA was used to study the difference between respiratory measures and degree of scoliosis grouped in degree 1, 2 vs. 3, 4 vs. 5, 6. Tuckey HSD was used as post-hoc test. All statistical analyses were performed using R version 4.0.1 (R Foundation for Statistical Computing, Vienna, Austria).

## 3. Results

### 3.1. Demographics of Participants

A total of 30 females (68%) and 14 males (32%) were included in the study with ages ranging between 13 and 51 years with an average age of 24.05 (9.09 SD). Most of the participants were Italian but other European nationalities were included (specifically, Turkish and Macedonian). Demographics data, including age at onset, age at diagnosis and information on the GAA triplet, are reported in Appendix A.

Age of onset of symptoms, as seen in Appendix A, ranged between 4 and 44 years old with a mean of 11.20 (6.67 SD), while age at genetic diagnosis was between 4 and 45 years with a mean of 15.36 (7.96 SD). Length of illness varied between 3 to 35 years with a mean of 13.00 (7.57 SD).

The first symptoms reported by patients in this study varied between scoliosis, some ill-defined motor dysfunction in infancy, lack of equilibrium or co-ordination in gait or cardiovascular abnormalities encountered during regular check-ups. Often patients encountered delays in genetic diagnosis likely secondary to the rarity of the disease or were diagnosed as a genetic screen of otherwise healthy relatives of patients with already confirmed FRDA. However, for the only group of siblings included in the study (subjects 1–2, 12–13 and 31–32), symptoms had presented for all three of them prior to genetic testing.

Age of onset at diagnosis is a recognized prognostic factor and five different patterns are commonly seen in FRDA patients, as described by Rummey et al.: (1) early onset, (2) typical onset, (3) intermediate onset, (4) LOFA (late onset) and (5) VLOFA (very late onset) [6].

GAA triplet expansions were also recorded and analyzed in the study. The GAA1 triplets number varied between 200 to 913 with a mean of 653.90 (180.70 SD) and GAA2 triplets ranged between 200 to 1436 with a mean of 845.13 (270 SD).

GAA triplet expansion inversely correlates with onset of disease and severity: the higher the number of triplets the earlier the disease onset [2]. This is also seen in our data, as early onset patients had a mean GAA1 triplet expansion of 695.25 (450–900, 148.63 SD) which decreased to 645.8 (200–900, 197.87 SD) in typical onset and 611.28 (449–839) in intermediate onset, ultimately falling to 400 for the LOFA patient.

### 3.2. Respiratory Measures

Forced vital capacity (FVC) data were adjusted for sex, age, weight and height. As shown in Table 1, most patients had FVC below the predicted for their demographics. In a restrictive respiratory pattern, an FVC less than 80% indicates mild insufficiency, which becomes moderately severe below 60% of predicted FVC and below 50% indicates inefficient cough and inability to protect the airway [9]. Furthermore, posture has an effect on FVC, as it modifies respiratory mechanics. In healthy subjects, a 7.5–10% reduction is noted in FVC from sitting to supine, which, if increased to 50–60%, suggests diaphragm weakness [10]. Interestingly, in FRDA patients this difference is less, and often, better FVC is noted when supine. This could be explained by truncal instability which improves when lying supine. Differences between FVC sitting and supine are represented in the following graph (Appendix A).

### 3.3. FEV1 and Tiffeneau Index 

FEV1 (forced expiratory volume in 1 s) in normal subjects corresponds to at least 80% of FVC, and it describes the strength of expiratory muscles. When analyzed together with FVC, it allows for differentiation via the Tiffeneau index between obstructive and restrictive patterns. Our results show a decreased FEV1 and Tiffeneau index of more than 80%, indicating that a restrictive pattern was common in our study population.

PEF values in healthy individuals usually vary between 600 and 1200 L/min, while PCF values below 180 L/min are a sign that cough efforts are not effective in the absence of respiratory pathology. The minimal value rises to 270 L/min in the presence of respiratory pathology [11]. In our population, PEF is more compromised than PCF with 83.7% of patients below the minimal level of 400 L/min. MIP and MEP values have no agreed reference value in healthy individuals. Values of MIP < 30 cm H_2_0 require assisted ventilation and MEP < 50 cm H_2_0 are a signal of inefficient cough. During the study design, the cut-off value of 80 cm H_2_0 was chosen, based on the recommendations of the American Thoracic Society and the European Respiratory Society [8].

In our study population, 80% of participants were below the minimal values for MEP and 66.7% for MIP, but if the clinical cut-off is considered, we note that cough may be inefficient for 15 out of 44 patients). The datasets for MIP and MEP were not complete (39/44), thus, while the values are reported in Table 2, statistical analysis of these values was not performed.

### 3.4. Overnight Oximetry and Capillary Blood Gas Analysis

Complete overnight oximetry data were available for 32 subjects (73%) (Table 3). FRDA patients often report daytime somnolence and fatigability. From our data, seven patients had an ODI > 5/h of which six could be classified as mild OSA (5–15 ODI) and one as severe (>30/h). The recorded ODI events show a minimal saturation reading of 75% but average saturations are not indicative of respiratory insufficiency in any of the participants. CO_2_ remained within normal levels in all participants.

In summary, in patients with FRDA there is sometimes a global reduction in FVC and FEV1. A reduction in FVC with a Tiffeneau index of 1 points towards a restrictive picture where expiratory function is more affected than inspiratory. Cough is partially effective, there is glottic incompetency and mild sleep-disorder-breathing is common but not associated with respiratory failure.

### 3.5. Motor Assessment

The study population had high SARA and FARS scores, highlighting a marked muscle weakness where abdominal muscles have a mean MRC score of 3.22, equivalent to weakness of muscle contraction against resistance, as well as low stability when standing (Table 4). This latter result can be explained by the majority of patients being completely or partially unable to ambulate.

### 3.6. Speech and Language Assessment

In restrictive pulmonary pathologies there is a decrease in FVC, and expiratory flow might be insufficient. In neuromuscular pathologies characterized by ataxia, lack of co-ordination may lead to dysarthria, speech tremor, intelligibility and increased inspiratory pauses [12].

In our analysis for speech and language the following items were taken into consideration:Numbers of syllables for expiratory phase (reading duration): normal value 21.71 syllables per expiration, SD: 5.38.Speed of speech (syllables per second including pauses): normal value 4.57 syllables per second, SD: 0.53.Speed of articulation (number of syllables excluding pauses in second): normal value 5.41 syllables per second, SD 0.62.Maximum phonation time (ability to maintain speech and duration of vocalization): normal 15–25 s, good 11–14 s, average 6–10 s and low 1–5 s. This is an indication of laryngeal and vocal cords’ efficiency.Expiration duration: normal 20–30 s, good 15–19 s, average 10–14 s and low 1–9 s. This is an indication of respiratory muscles efficiency in relation to contractility and co-ordination.

For one patient, speech and language assessment was not completed, thus the dataset includes 43 records. The mean results are shown in the Table 5.

### 3.7. Scoliosis Assessment

In our analysis we included 32 patients (73%) all of whom had x-ray spine in anterior-posterior views. For the twelve other patients imaging was not available. The majority of patients presented with right thoracic deviation; in 13, a double curve could be seen in the X-ray: a thoracic and a lumbar one. Kyphosis data are unfortunately not available to us due to lack of raw data. Degree of scoliosis was classified based on Cobb degrees: 12.5% of patients had curves less than 20° (score 1), 32% had curves between 20 and 40° (score 2 or 3) and 56.5% presented curves angles above 40° (score 4, 5, 6 or 7 for which surgery is an indication).

### 3.8. Statistical Correlation of Results

FVC, FEV1 and MEF inversely correlate (*p* < 0.001, Spearman’s ρ) with the severity of disease as expressed by SARA in upright and supine positions. For absolute values significant findings can only be found for SARA, however, the majority of respiratory parameters remains below average.

Similar results are obtained for the kinesiologic measures (abdominal muscle strength and upper limbs co-ordination), showing a significant correlation with the percentage of predicted FVC (*p* < 0.001, and *p* = 0.001, respectively, Spearman’s ρ).

No significant correlation was found between speech speed or speed of articulation and respiration parameters.

One-way ANOVA was used to study differences between spirometry results and scoliosis severity and the results show a statistically significant relationship between FVC (% predicted) and degree of scoliosis (*p*-values of 0.037 to 0.033, respectively). Post hoc test showed significant results between scoliosis severity group 1–2 and 3–4 (*p* = 0.0030 and 0.0036) but not between these and the most severe group 5–6.

Following these findings, we hypothesized that localization of curvature could have an effect on spirometry results, with thoracic curvatures having more of an effect on spirometry results. We therefore re-ran the analysis including only thoracic scoliosis. Here too, the differences were significant for the groups with less severe scoliosis but flattened for the last group with the most severe curvature (Table 6).

When considering only thoracic scoliosis, there is a more significant statistical relationship between degrees of scoliosis and FVC (*p*-values of 0.027 to 0.014) and a moderate relationship between FEV1 when supine and scoliosis (*p*-values of 0.042). Therefore, while scoliosis contributes to changes in spirometry results, it is not the main determinant.

## 4. Discussion

FRDA is a multisystem progressive pathology linked to the frataxin gene. It is characterized by central and peripheral ataxia, muscular weakness, skeletal deformities, cardiomyopathy, endocrine and psychiatric disturbance [2].

In our study we aimed at exploring respiratory function in FRDA patients by a systematic re-evaluation of clinical data accumulated across 3 years in our institution. This was a retrospective, observational study based on data collected between 2018 to 2021, from 44 subjects aged between 13 and 51 years old with a confirmed genetic diagnosis of FRDA. The observed data included: demographics of the participants, spirometry results, sleep studies, blood gas analysis, speech and language assessment and musculoskeletal assessment. Statistical analysis confirmed the presence of a recurrent restrictive respiratory pattern in FRDA patients with reduction in FVC, FEV1 and PEF; PCF, MEP and MIP were all below expected values, with more severe impairment observed in FEV1 (77.64% of predicted). Considering that over 70% of our patients had a disease duration of <15 years, this points to an early appearance of respiratory involvement in FRDA. Respiratory function assessment should be therefore considered for all FRDA patients irrespective of age or disease duration. However, the severity of restrictive pattern in spirometry findings statistically correlates with the severity of disease as measured with SARA with the global speech and language assessment and with degree of scoliosis, especially when considering the thoracic curve. The stronger correlation found with SARA is explained by the higher granularity of the scale compared with the coarse categorical scale of FARS stages of disease.

From our study we have seen how FRDA, which leads to loss of co-ordination and muscle strength, also affects diurnal and nocturnal respiratory efficiency. The most compromised parameters are those of the expiratory phase and they are mainly influenced by abdominal muscle weakness, lack of upper limb co-ordination and glottic muscle co-ordination (in particular, the ability to maintain phonation during expiration). By looking at the pulmonary function tests and the correlation with the disease severity, we see that expiratory function worsens more than inspiratory with more direct correlation with disease.

Contrary to what is usually observed, we recorded an improvement of the respiration performance in the supine position compared to the sitting or standing position [13]. This might be linked to the additional effort the ataxic patient requires to stabilize the trunk while sitting.

In conjunction with difficulty in maintaining an erect posture secondary to skeletal deformities and dysphagia, we believe that respiratory pattern changes and cough inefficiency are clinical findings that need to be addressed via a rehabilitation programme in order to minimize their progression and promote adaptation. Dysphagia and dysdiadochokinesia, frequent and characteristic signs of FRDA were not formally assessed in this study, but might also have contributed to the difficulties seen in the erect posture.

Only one of our patients was under nocturnal NIV, prescribed for symptomatic OSA. Even though none of the other participants showed signs of nighttime respiratory insufficiency, many of them had desaturation events of significant duration. It might be worth considering a more systematic appraisal of the nighttime respiration efficiency and the possible subtle daytime signs and symptoms of OSA and its effects on the ability of patients to perform activities of daily living.

The speech and language assessment shows more severe impairments in parameters related to co-ordination. The difference between speed of speech and articulation speed showed how many pauses the patient required in order to articulate speech which could be secondary to inco-ordination between the action of speech and breathing [14].

When looking for studies published in the literature on respiratory function in patients with FRDA, they are few and often dated. In a multicentre study by Tsou and coauthors in 2011, the causes of mortality in 61 individuals with FRDA between 1993 and 2010 were analyzed. Of these, 59% died from cardiac causes while 9.8% died from pulmonary causes [15]. Thus, the respiratory impairment while relevant as concurrent morbidity does not seem to represent a primary cause of mortality. The longitudinal follow-up of our cohort may further clarify this issue.

In 1979 Begin at al. studied respiratory mechanics in 11 patients with mild to moderate forms of FRDA. They showed that central dysregulation does not play a role in respiratory dysfunction in patients with FRDA. They hypothesized that restrictive respiratory pictures were caused by scoliosis [16]. On the other hand, in 1976, spirometry data were collected from 20 young FRDA patients by Bureau et al. which showed how the restrictive respiratory compromise was caused by neuromuscular impairment rather than thoracic deformity [17]. Our data support the prominent role of in-coordination and muscle weakness in modulating respiratory efficiency, with scoliosis weighting as a secondary conditioning element, especially when considering the thoracic curve. Therefore, while a direct central mechanism may not be the primary cause of the respiratory dysfunction, the neuromuscular impairment associated with the central nervous system damages characterizing FRDA plays a central role.

Respiratory impairment is described in other inherited ataxias. In ataxia-telangiectasia (AT), the restrictive pattern with reduced FVC is complicated by the recurrent infections, and respiratory compromise is one of the main morbidity indicators [18]. In patients with spino cerebellar ataxia (SCA) type 1, 2 and 3, the profile is subtle and includes both restrictive and obstructive pathology [19]. Few studies report on the rehabilitation approach for this specific problem.

One study reported positive results in 11 AT patients following a 24-week training program consisting of 20 min daily exercise at 60% of MIP [20]. The positive results were recorded not only on the specific indicators of respiratory function, but also on subjective indicators of dyspnea and on more general measures of quality of life, underscoring the importance that respiratory wellbeing has on health as a general construct.

### 4.1. Strategies for Treatment

There are no established guidelines or recommendations for rehabilitation of the respiratory function in FRDA, but the experience obtained in other neuromuscular disorders [21] may guide the rehabilitation program. From the results of our study and from the experience accumulated in these years there are some lines of action that are both logical and effective, at least as assessed by clinical judgment. Rehabilitation sessions include work to improve the self-awareness of respiration and thoraco-abdominal co-ordination, exercises to improve respiratory muscle strength through volume incentivators for the inspiratory phase such as the Coach-2 and expiration trainer such as the Threshold-PEP. Great attention is also given to the lumbar and abdominal strength, stabilization and control, using an elastic band if needed. When cough is inefficient, we usually introduce exercises and instruct the caregivers for maneuvers facilitating expectoration and for the use of the Cough Assist. Each rehabilitation program needs to be carefully drafted upon the individual needs and the function impairment emerging from the systematic evaluation.

### 4.2. Limitations and Strengths of the Study

The size of our participants’ sample, albeit not large is still comparatively one of the most numerous in the literature on the topic. Furthermore, having the ability to carry out spirometry assessment in house with the help of a specialized respiratory physiotherapist and shared speech and language therapy protocols allowed for uniformity in data collection. Some unexpected findings such as the decline in respiratory performance seen in most subjects in the erect posture, could be due to technical errors during spirometry or variability in collaboration and understanding instructions while recording the spirometry measures. A re-test confirmation as well as a longitudinal follow up could confirm the finding. A 2:1 female prevalence is observed in our cohort, in spite of gender ratio of 1:1 reported in the literature [2]. This skewed gender representation is casual but should be considered, since it might affect the results. For example, COPD manifestations differ between males and females with the latter being more at risk of developing severe disease [22]; similarly respiratory impairment in FRDA might impact differently in the two genders, however this will require further studies.

Unfortunately, due to the retrospective nature of the study, some of the data were not available. This was true for speech and language, for overnight oximetry and for radiological imaging of scoliosis. We utilized desaturation index as a surrogate marker for OSA; a more direct measure such as that provided by polysomnography should be added to achieve a secure diagnosis. Another weakness of the study is its cross-observational nature, lacking prospective evaluations and thus information on the progression of the observed impairment. We plan to follow up our FRDA cohort with at least yearly checks, thus filling this gap. Besides function indices, it would also be interesting to provide measures of activity limitations, thus weighting the impact of the respiratory problems on the patients’ general functioning.

## 5. Conclusions

Based on the results obtained from this study, we believe that standards of care for FRDA patients should also take into consideration changes in respiratory function, cough inefficiency and possible nocturnal hypoventilation. All FRDA patients should have a spirometry assessment and be offered rehabilitation treatment if compromise is found. Treatment should include respiratory physiotherapy aimed at improving respiratory dynamics and vital capacity, global and selective muscle weakness, posture education and speech and language therapy. Treatment for scoliosis should be carefully planned and regular evaluation of respiratory function ought to be continued to ensure those patients who require respiratory support can be readily identified.

## Figures and Tables

**Table 1 children-09-01319-t001:** FVC in FRDA patients.

Spirometry	Mean (SD)	Range
FVC sitting % predicted FVC	2.84 L (0.86)	1.4:5.4
	74.83% (18.21)	36.6:105.5
FVC supine % predicted FVC supine	2.84 L (0.79)	1.3:5.3
	74.72 %(16.67)	33.6:105.5
% difference between supine and sitting	−0.74 L (9.29)	−36.5–14.77

**Table 2 children-09-01319-t002:** Spirometry results: FEV1 sitting and supine, Tiffeneau index and PEF (peak expiratory flow), PCF (peak cough flow), MIP (maximal inspiratory pressure), and MEP (maximal expiratory pressure) values. The study population was subdivided in subjects above and below a cut-off value indicating healthy individuals.

Spirometry	Mean	Range
FEV1 sitting	2.58 L (0.77)	1.2; 4.8
% of predicted FEV1 sitting	77.64 (18.77)	36; 110.8
FEV1 supine	2.46 L (0.77)	0.1; 4.5
% of predicted FEV1 supine	75.80 (15.95)	31.7; 105.9
Tiffeneau index sitting	88.13 (14.82)	0; 104.3
<1, *n* (%)	1 (2.3)	
1+, *n* (%)	43 (97.7)	
Tiffeneau index supine	84.17 (18.75)	0; 97.7
<1, *n* (%)	1 (2.3)	
1+, *n* (%)	43 (97.7)	
PEF	310.81 (114.39)	130; 710
<400, *n* (%)	36 (83.7)	
400+, *n* (%)	7 (16.3)	
PCF	365.47 (106.66)	175; 670
<270, *n* (%)	9 (20.9)	
270+, *n* (%)	34 (79.1)	
PCF/PEF	1.23 (0.17)	0.4; 1.1
>1, *n* (%)	36 (83.7)	
<1, *n* (%)	7 (16.3)	
MIP	70.23 (23.05)	30; 141
<80, *n* (%)	26 (66.7)	
80+, n (%)	13 (33.3)	
MEP	64.08 (27.29)	29; 163
<80, *n* (%)	32 (80.0)	
80+, *n* (%)	8 (20.0)	

**Table 3 children-09-01319-t003:** Overnight pulse oximetry and capillary PCO2 readings.

Overnight Oximetry	Mean (SD)	Range
Index	4.77 (6.79)	0.5; 38.2
Artifacts	1.22 (1.11)	0.1; 4.9
Resting SpO2	95.87 (4.93)	69.9; 98
<88% SpO2 (min)	1.30 (2.09)	0; 8
SpO2 minimum value	82.90 (6.87)	64; 92
Average SpO2	92.72 (1.07)	89.9; 93.9
PCO2	36.43 (3.45)	30; 42.7

**Table 4 children-09-01319-t004:** Motor assessment.

Motor Function	Median (SD)	Range
SARA scale	23.00 (8.16)	8–34
Abdominal strength as MRC	3.00 (1.41)	1–5
Upright stability as FARS	24.25 (6.31)	1.5–28
Upper limb co-ordination as FARS	11.50 (6.51)	2–36
FARS functional staging	4.00 (1.12)	1.5–5

**Table 5 children-09-01319-t005:** Speech and language analysis.

Speech and Language Analysis	Mean	Range
Reading duration	14.99 s (5.96)	5.5–35.3
Speed of speech	3.29/s (0.84)	1–5
Articulation speed	4.33/s (1.37)	1.8–11.3
Max phonation time	11.79 s (4.85)	3–25
Expiration duration	11.05 s (5.78)	3–27

Speech and language analysis identified difficulty in speech speed and articulation.

**Table 6 children-09-01319-t006:** One-way ANOVA: relationship between spirometry and thoracic scoliosis.

Variable	Statistic	*p* Value
FVC-sitting	4.64	0.019
FVC sitting %predicted	4.14	0.027
FVC supine	5.04	0.014
FVC supine %predicted	4.15	0.027
FEV1 sitting	3.02	0.065
FEV1 sitting %predicted	2.93	0.071
FEV1 supine	3.57	0.042
FEV1 sup %predicted	2.76	0.081
TIF sitting	0.68	0.517
TIF sitting	0.30	0.746
PEF	0.52	0.601
PCF	1.40	0.265
PEF/PCF	0.38	0.690
MIP	0.67	0.519
MEP	1.02	0.376

## Data Availability

Data supporting reported results are stored in the institutional servers and might be shared in anonymized form upon motivated request.

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
