# Peer review of "Respiratory Function in Friedreich’s Ataxia"

_children, 2022, doi:10.3390/children9091319_

Round 1

Reviewer 1 Report

Manuscript title: Respiratory Function in Friedreich’s ataxia

This study aimed to provide a picture of the respiratory function of patients with FRDA. The manuscript is original, scientifically sound and in consonance with the Journal’s scope. However, the description of the methodology and results needs to be improved. Below you will find some comments intended to improve the manuscript.

Introduction

  • Respiratory compromise - change to respiratory impairment 

  •  6 minutes walking - change to 6-minute walk test

  •  day-time somnolence - change to day-time sleepiness

  • respiratory difficulties - change to respiratory impairment

Materials and Methods

  • Authors should state the complete study setting. Please add the city and country after “IRCCS Eugenio Medea “ La Nostra Famiglia”.

  • In line 97 “For the respiratory assessment (performed between 2018-2021)” it is not necessary to repeat “(performed between 2018-2021)”.

  • Line 98. In case of NIV use, which method was used?

  • Line 100 authors should declare if they followed ATS/ETS recommendations

  • Add which variables of spirometry were considered. Forced vital capacity and forced expiratory volume in 1s should be stated here. Same for MIP and MEP.

  • “For speech assessment duration of phonation (number of syllables per expiratory phase, length of vocalization per second), speed of speech (numbers of syllables per sec- 112 ond) and duration of expiration were included.” - please specify the methods used for speech assessment (recordings? Electronic health records? softwares involved?)

  • Line 115. “legibility” - do you mean intelligibility?

  • Line 116. Please improve statistical description. The ANOVA and post-hoc tests are not used to identify correlation. You should also add the software used to the statistical analyses and the level of significance considered.

Results

  • Line 120. Add female/male percentages

  • Line 122l. “but other European nationalities were included” - which ones?

  • Table 1 - it would be clear to the readers to see at the end of this table the average values for Age, Age at onset of symptoms, Age at diagnosis and Length of illness from diagnosis to study.

  • Add legend to Table 1

  • Line 133. “for the only group of siblings included in the study” - please describe the individuals’ ID, according to Table 1.

  • Line 136. Correct the typo

  • Line 149. It is not necessary to repeat this information “Forced vital capacity (FVC) was recorded in all patients sitting and supine.”

  • Line 152. Please add a reference to support your statement: “In a restrictive respiratory pattern, an FVC less than 80% indicates mild in- 152 sufficiency, which becomes moderately severe below 60% of predicted FVC and below 153 50% indicates inefficient cough and inability to protect the airway”

  • Line 155. Same suggestion

  • Line 157. “Interestingly in FRDA patients this difference is less and often, bet- 157 ter FVC is noted when supine.” - Normally this is caused by technical errors during the spirometry. It could also reflect a lack of understanding of the assessor's instructions by the patient. Authors should add this finding to the study limitations.

  • Line 158 please add a reference

  • Figures - it would be better to see figures showing average, maximum and minimum values for FVC, FEV1 (for example) and not the values for each individual separated.

  • Table 2. in the rows “FVC sitting” and “FVC supine” please add (volume L) or just add L after the mean values.

  • Line 170 correct the typo

  • Lines 168-170 this is part of the methods section

  • Peak Expiratory Flow and Peak cough flow should be firstly mentioned in the methods section. 

  • Lines 173-175 this excerpt is part of the methods section

  • Line 183-186. “MIP and MEP values are unfortunately incomplete as 185 analyses of these values were not performed.”. Instead of stating this, I would suggest informing the readers that there was a lack of complete datasets for these variables and state the available N (39/44). This suggestion is because you actually included these values on Table 3. 

  • Table 4. Please correct SpO2 and PCO2 by changing “0” (zero) to “O”

  • Table 4. “<88% Sp02 after 1 minutes” - did you mean <88% Sp02 (minutes)?

  • Lines 204-212. This is part of the methods section. Please consider this suggestion to all following similar sub-headings of the results section.

  • Line 219. “Phonological assessment” - change to “Speech and language assessment” or “speech and language analysis”, the term you used in Table 6.

  • Line 232. “Maximum speech time” - Did you mean “maximum phonation time”?

  • Line 235 - “Length of expiration (ability to sustain speech during expiration, expiration duration)” - this variable description is conflicting with the previous one. It seems inaccurate. Just “expiration duration” seems more appropriate.

  •  Table 6. “Reading duration”

  • Lines 240-243 The interpretation of the results should be moved to the discussion section.

  • Line 245 - “In our analysis we included 32 patients” - state the (%)

  • Line 256 - “we can deduct that expiratory func- 256 tion worsens more than inspiratory with more direct correlation with disease.” - discussion

  • Line 262 - “lying position” - please change to supine position

  • “This might 262 be linked to the additional effort the ataxic patient requires to stabilize the trunk while 263 sitting.” - This is also part of the discussion. Please follow this suggestion for all subsequent similar sentences interpreting results.

  • Table 7. Replace the software variable names for text (e.g. FVC_sitting_% predicted).

  • Table 7 One-way ANOVA spirometry and degree of correlation. Which statistical information exactly expresses the “Correlation” column? Is this the ANOVA’s F-test? Which groups were compared? All this information should be stated in the methods section.

  • Post-hoc tests are absent at all in the manuscript. Please add the results or remove them from the methods.

Discussion

  • Line 284 - add references

  • Lines 286-292. This paragraph is not necessary because this information is stated previously in the manuscript.

  • Line 304 - needs to be reworded

  • “In conjunction 308 with difficulty in maintaining an erect posture secondary to skeletal deformities and dys- 309 phagia we believe that respiratory pattern changes and cough inefficiency are clinical 310 findings that need to be addressed via a rehabilitation programme” - Swallowing assessment was foreseen in the methods but not in the results. The same applies to dysdiadochokinesia assessment.

  • Line 313. add references

  • 318. reword “doesn’t”

  • Line 339-340 - “for one patient speech and language assessment was not done” - lack of complete datasets should be stated on the results section

  • I would suggest adding a subheading in the discussion with strategies to tackle the respiratory dysfunction in patients with FRDA (e.g. breath stacking, inspiratory/expiratory muscle training, respiratory muscle endurance training, cough assist, NIV).

  • You cited non-invasive ventilation in Line 99  but at no other time do you address this issue, which is very important.

  • In general I feel that the authors have included too few references. It is understandable that there are few papers on FRDA because it is a rare disease. But the authors could expand the bibliography in the discussion by including papers on respiratory function in other neuromuscular diseases (e.g. relation of respiratory function to speech and swallowing).

Limitations

  • Limitations. “The size of our participants' sample is one of the strengths of our study as it allows, 333 even in the case of a rare disease, to run statistical analysis.” - if you opt to keep this sentence here, I suggest adding a statistical power analysis.

  • Overnight desaturation index is a surrogate marker for diagnosis of OSA. This should be stated here.

Conclusions

  • The authors' conclusion seems appropriate and in line with the results obtained. However, it is not corresponding with the objectives of the study. The authors may choose to improve either the study objectives or the conclusion. Also, the conclusion on the abstract is not consistent with the manuscript's main conclusion.

Abstract

  • Line 16. “44 subjects between 15 and 51 years” is not consistent with the results section.

Author Response

The point per point responses to the reviewer's comments are given in the attached file

Reviewer 2 Report

This report will be helpful and informative to clinicians. I like it.

Please make sure the clinical trial in the report was properly registered. 

Make sure it meets specific criteria for content, quality and validity, accessibility, etc,.

Author Response

We thank the reviewer for his appreciation. The study was retrospective and thus while evaluated by the ethics committee was not recorded as a clinical trial. We trast the criteria mentioned are fully respected. 

Reviewer 3 Report

In the current paper, the authors retrospectively analyzed respiratory function in 44 patients suffering from Friedreich’s ataxia. The investigated respiratory core values like FVC, FEV1, PEF, PCF, MEP, MIP as well as ODI in FRDA patients and showed that a predominantly restrictive respiratory impairment can be found in FRDA patients.

1.)    Introduction

Some small minor typos can be found – e.g.  line 32 Friedreichs ataxia > should be Friedreich’s ataxia. I would recommend to re-phrase the first sentence of the second paragraph (line 37-38).  I am not familiar with the terms “central and peripheral” ataxia – one can imagine what is meant but the proper terms would be cerebellar or spinal ataxia.

The second part of the introduction is for my taste a bit too long and could be shortened or incorporated into the discussion part later.

2.)    Materials and Methods

The materials and methods are described sufficiently, although the statistical analysis is with just one sentence quite short (p-value is missing, what kind of post-hoc tests, which statistical program has been used). There more information has to be provided.

3.)    Results

3.1 Demographics of participants: I would suggest to put Table 1 as supplementary data. Strikingly there is a female predominance with a ratio of female: male of 2:1 – maybe the authors could comment on that, especially if there is an apparent gender difference in respiratory function.

3.2 Respiratory function: Figure 2 does not really give more information and could be skipped.

3.2 > should be 3.3 FEV1 and Tiffeneau index: Some abbreviations are NOT explained in the text and just in the table3 below (i.e. PEF, PCF, MIP and MEP and ODI in line 193) – please explain them in the text. One typo can be found on line 170 – should read FVC and not VFC. In line 173 it should read 600 and 1200 L/min. The percentages mentioned in the text are wrong and have been swapped – in line 176-177. The sentence “PEF is more comprised than PCF with 83.7% of patients below the minimal level of 270L/min“ is misleading. According to table 3, PEF was < 400 L/min, and PCF was < 270L/min. So please correct this statement.

3.4 Motor assessment: Since SARA and MRC as well as FARS are strictly taken not metric but ordinal, the median and range would be more appropriate.

3.7 Statistical correlation of results: this first paragraph is not quite clear to me. First the authors claim that FVC, FEV1 and MEF (supposed to be MEP?) inversely correlate with SARA and FARS scores. But then for absolute values significant values can only be found for SARA – please explain what is meant with that.

The “correlation” between spirometry results and scoliosis, especially Table 7 and 8 is misleading – what is the difference between the 2 tables? Use one of them but not both – was there an appropriate statistical approach for re-running the analysis in order to get more significant values?

A linear/multiple regression model might be a more suitable and appropriate statistical method.

4.)     Discussion and 5.) Conclusions

Nothing to comment on.  

Author Response

The point per point responses to the reviewer's criticism are detailed in the attached file

Round 2

Reviewer 1 Report

The authors met all my requests. Table S1 is absent from the supplementary material and should be added.

Best regards!